# Hazardous Solid Waste Confined in Closed Dump of Morelia: An Urgent Environmental Liability to Attend in Developing Countries

**M. Lourdes González-Arqueros** [1], **Gabriela Domínguez-Vázquez** [2], **Ruth Alfaro-Cuevas-Villanueva** [3], **Isabel Israde-Alcántara** [3] **and Otoniel Buenrostro-Delgado** [3],*

1   CONACYT, Instituto de Investigaciones en Ciencias de la Tierra, Universidad Michoacana de San Nicolás de Hidalgo, Morelia 58000, Mexico; lourdes.gonar@gmail.com
2   Facultad de Biología, Universidad Michoacana de San Nicolás de Hidalgo, Morelia 58000, Mexico; gdoguez@yahoo.com.mx
3   Instituto de Investigaciones en Ciencias de la Tierra, Universidad Michoacana de San Nicolás de Hidalgo, Morelia 58000, Mexico; ruth.alfaro@umich.mx (R.A.-C.-V.); isabel.israde@umich.mx (I.I.-A.)
*   Correspondence: otoniel.buenrostro@umich.mx; Tel.: +52-443-325-6301

**Abstract:** In developing countries, landfills of urban solid waste (USW) are a major source of contamination. One reason is the common practice of the illegal confinement of hazardous waste (HW). The contamination is mainly due to deficitary design location, operation and lack of liner, which enables the dispersion of pollutants. The aim of our work is to demonstrate the presence of heavy metals (HM) and arsenic (As) in USW of the closed dump of Morelia, which clandestinely confined HW for 20 years. Solid samples of USW were collected from eight opencast wells with different age of confinement. Composition, degradation status, physical-chemical characterization and analysis of HM and As were carried out. The results showed the presence of Pb, Cu, Ni, Zn, Cr, Fe and high concentrations of As. This study provides evidence about the usual and illegal practice of landfill HW together with USW; the hazard due to the presence of HM and As; the deficiency in the operation and closure; and, the lack of competent legislation on the subject. This information is essential to establish background information for improving laws and help decision makers in territorial planning to improve public and environment health.

**Keywords:** metals; arsenic; pollution; Mexico; developing countries; landfill; urban solid waste; disposal; waste management

## 1. Introduction

Environmental pollution is a consequence of the irrational use of natural resources. In developing countries, the sites for disposal of urban solid waste (USW) are a major source of contamination [1] where, through leachates, pollutants can be dispersed to soil and groundwater [2,3]. This contamination results from the deficiency in the design and operation of the landfills in which the USW are deposited, as well as the lack of monitoring of closed dumps and scarcity of environmental legislation [4].

Typically, closed dumps are considered environmental liabilities, as well as geographic sites polluted by the release of materials. Some of the most common landfills pollutants are heavy metals (HM). Very often, the effect of the pollutants is increased when the landfill is closed; since it is a common practice of the use of closed dumps as farmlands in urban and sub-urban centers in developing countries [5,6].

Accelerated urbanization in developing countries, which has affected the urban metabolism of human settlements, has generated a problem that is reflected in urban and environmental functionality [7]. In general, the so-called landfills of developing countries and which more closely resemble the category expressed in [4] as open dumps or controlled dumps, are located in areas of high marginalization, characterized by low

per capita income [2,8] and high population density, in addition to not complying with environmental legislation operating as dump sites.

The northwest of Morelia city, where the closed landfill is located, is no longer a predominantly rural area and has become a densely populated human settlement with low economic income, mixed with agricultural and forested areas. The conurbation of this area with the closed dump and the landfill results in greater exposure and vulnerability to highly polluting and dangerous particles for the population health. In parallel, this site is located in a fracture zone, with highly porous soils and in a groundwater recharge area.

Previous work at the study site [9] reported high concentrations of cadmium, nickel, arsenic, lead, hexavalent chromium, and total chromium in leachates. The central problem is that there should not be heavy metals in USW. However, the confinement of hazardous waste (HW) in these landfills has become a common practice in many emergent nations. Therefore, the concentration of heavy metals in USW and the toxicity of them depend on the amount of hazardous waste it contains [10].

The aim of our work is to demonstrate the presence of heavy metals in the USW, clandestinely confined together with hazardous waste due to non-compliance with current environmental legislation. In this way, our study provides current data on the increasing environmental and public health risk represented by closed dumps or controlled dumps in developing countries.

## 2. Materials and Methods

### 2.1. Physical Description of the Study Area

The study area (~128 km$^2$) (Figure 1) is located in Morelia, Michoacán, Mexico, in the Trans-Mexican Volcanic Belt, within the Michoacán-Guanajuato volcanic complex. The area lies in a recharge site of an overexploited aquifer that provides water to more than 120,000 people [11].

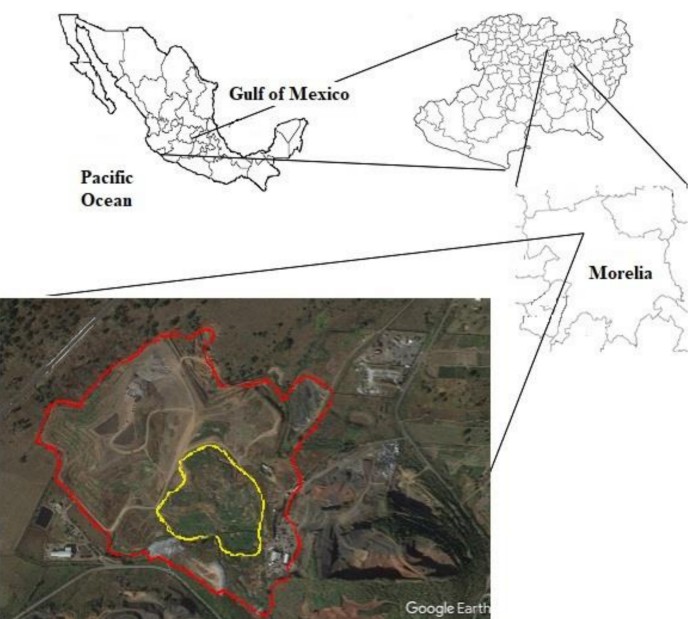

**Figure 1.** Location of the study area (Modified from Google Earth 2017$^®$; INEGI). The red dotted line indicates the total perimeter dedicated to the landfill. The yellow dotted line indicates the closed dump area where the samples were collected.

The area presents two important hydrogeological conditions since it is located on permeable geological materials and is affected by a regional fault system. Two semi-shield volcanic bodies dominate the landscape in the northeast and in the southwest. The volcanism and the faults determine the relief, with dominant E–W structures, associated

with the Morelia-Acambay fault system [12], in which two faults are notorious because of their size, "El Cerrito" and "Cointzio".

The closed dump of Morelia was built without liner or structure engineered and designed to control the percolation and infiltration of leachates and biogas generated in the site [9]. The site was in operation as an open-air dump since 1997 until 2007, in which an average of 900 tons of USW per day was deposited. The closure of the closed dump was carried out from the leveling and compaction of the solid waste, followed by a covering with a layer of approximately 10 cm of clay and tezontle (volcanic rock).

There is evidence that along with USW, hazardous waste was deposited illegally. Through visual identification of piles, open pits, and the type of waste at the time of truck discharge [13], 18 different sources of HW were recognized (Table 1); the varied list shows from inert (sharp objects) to organisms waste (animals and humans). The most frequent generators were medical offices 25%, automotive maintenance shops 21%, construction 14%, industries and garages 12%, and the remaining groups 28%. Surprisingly, waste from paper manufacturing and solids from wastewater treatment continued to be deposited even when the site was closed.

**Table 1.** Sources and type of waste identified in the illegal disposal of hazardous waste in the Morelia closed landfill. Source: [13].

| Source | Waste |
| --- | --- |
| Agricultural equipment stores | Plaguicide (pesticides, herbicides) <br> Fertilizers |
| Hospitals/clinics/doctor's offices/Clinical and radiological analysis laboratories | Sharp objects <br> Human waste <br> Healing material wastes <br> Syringes <br> Developing material/worn plates |
| Research laboratories | Reagent containers <br> Organisms remains |
| Veterinary clinics | Dead animals <br> Animals waste <br> Healing materials |
| Photographic developing workshops | Photographic and film wastes <br> Containers of developing products |
| Computer, photocopier and printer maintenance workshops | Ink cartridges <br> Accessories <br> Toner cartridges <br> Photocopier oil waste |
| Beauty salons <br> Slaughterhouses, butchers, chicken shops, guts dispensing | Beauty products <br> Dye residues <br> Animal waste |
| Housing | Household cleaners <br> Medicines and drugs <br> Cosmetics <br> Pests, garden waste and batteries |
| Sand mines/coal sale | Sand waste <br> Coal waste |
| Electrical workshops | Incandescent lamps <br> Light bulbs <br> Batteries |
| Paint stores | Paint containers <br> Paintbrushes <br> Containers of varnishes, thinner, turpentine, gasoline <br> Rag with solvents traces |
| Gas stations | Oil residues <br> Cleaning material residues |
| Hardware stores | Solvent containers |

**Table 1.** *Cont.*

| Source | Waste |
|---|---|
| Car parts stores | Oil containers<br>Used car parts<br>Antifreeze containers |
| Auto body shops | Paint containers<br>Car parts<br>Polisher containers |
| Garages | Oil containers<br>Tires<br>Used car parts<br>Inner tubes |
| Paper industry | Solids from wastewater treatment<br>Waste from paper manufacturing |
| Construction companies | Cement waste<br>Lime residues<br>Tile glue residues |

*2.2. Sampling*

The area of the closed dump was divided into four quadrants oriented from southwest to northeast (Figure 1). Eight sites were randomly selected, ensuring that half of the samples will have a confinement time of 5 years and the remaining half will have a confinement time of 10 years. Opencast wells were dug during the dry season to a depth of three meters with a backhoe loader with extension (Case 2002®). Subsequently, approximately three kilograms of solid samples of USW were taken. USW samples were placed in black polythene bags, labeled and placed in a cooler for their transfer to the laboratory. Inside the opencast well, the in-situ temperature was measured with a digital thermometer.

*2.3. Sample Characterization*

The solid samples of USW were characterized according to the Mexican Official Norm NMX-AA-022-1985 [14]. By-products were manually separated and grouped into two fractions: organic and inorganic. Then, the organic fraction was grouped into categories of degradability according to the classification proposed in [15].

Physicochemical analyses were performed according to the Mexican Official Norm NMX-AA-052-1985 [16]. The components of the samples were crushed with scissors and ground with an analytical mill (MF 10®) (with a one-millimeter sieve), deposited in plastic jars and frozen at $-4\,°C$. Subsequently physical-chemical parameters as moisture (NOM NMX-AA-016-1984) [17], pH (NOM NMX-AA-25-1984) [18], total dissolved solids (TDS) NMX-AA-016-1984 [19], and volatile solids (VS) (2540G technique from Standard Methods) [20] were determined.

The concentration of heavy metals was determined from one gram of aliquot of each sample by acid digestion of sediments with a flame atomic absorption spectrophotometer (FLAA), according to the EPA method 3050B [21]. Arsenic was determined with the hydride generation method according to the Mexican Official Norm NMX-AA-051-SCFI-2016 [22]. Analyses were performed in duplicate.

In order to analyze the presence of significant differences among the metals concentrations and in the content of the organic fraction according to the confinement time of the USW, the results were captured in a database and processed with descriptive statistics and analysis of variance (ANOVA) using JMP 8 software.

**3. Results**

The organic fraction percentages per sample ranged from 48% to 67%. An average of 54% of the sample was the organic fraction and 46% the inorganic fraction. The categorization of the USW organic fraction samples (Table 2) showed that 82% of the by-products were of very rapid degradation since they mainly derived from food residues. Of note, 13%

of the samples showed moderately slow and slow degradation; this type of degradation is associated with the fraction of residues that, although organic, have a higher content of cellulose and lignin compared to those that come from food residues.

**Table 2.** Degradability of the organic fraction of the solid residues (% fresh weight).

| Degradability of the Organic Fraction | % |
|---|---|
| Very rapid | 82.0 |
| Moderately rapid | 4.0 |
| Moderately slow | 10.6 |
| Slow | 2.3 |

The physical-chemical characterization (Table 3) showed statistically significant differences between temperatures for both confinement periods. The predominantly basic pH values, together with the low moisture content, directly influenced the rate of degradation of the organic matter. Total solids (TS) values varied between 58% and 78%, with an average of ~68%. These values are considered high regardless of the USW confinement time. On the other hand, the VS values showed a wide variation, between 17% and 79%, and the averages of the samples according to confinement time did show significant differences. Likewise, the values of the ash, the values of VS, also showed a high variability and significant differences between averages. It reaffirms the previous results and corroborates a high variation in the degradation state of the organic fraction of solid waste within the study site.

**Table 3.** Average values of the physicochemical parameters of the urban solid waste (USW) with 5 and 10 years of confinement.

| Parameter | Average ± Standard Error | Years of Confinement | Significance |
|---|---|---|---|
| Temperature (°C) | 26.5 ± 0.44 | 5 | * |
| | 35.0 ± 0.86 | 10 | * |
| pH | 8.35 ± 0.03 | 5 | NS |
| | 8.14 ± 0.14 | 10 | NS |
| Moisture (%) | 31.7 ± 1.66 | 5 | NS |
| | 31.8 ± 1.24 | 10 | NS |
| Total solids (TS) (%) | 68.2 ± 1.66 | 5 | NS |
| | 68.1 ± 1.24 | 10 | NS |
| Volatile solids (VS) (%) | 73.1 ± 2.86 | 5 | * |
| | 52.1 ± 3.65 | 10 | * |
| Ash (%) | 26.8 ± 2.86 | 5 | * |
| | 49.4 ± 3.41 | 10 | * |

* = Significant. NS = Non-significant.

The heavy metals in the solid residues of the closed dump were lead (Pb), copper (Cu), nickel (Ni), zinc (Zn), chromium (Cr), iron (Fe), and the arsenic metalloid (As) (Table 4). The Pb values for samples 3, 6 and 7 were substantially higher compared to the rest (89, 149.67 and 108.67 mg/kg, respectively). The Cu values for sample 3 also showed an exceptionally higher value (744.17 mg/kg) compared to the rest of the samples. This sample also presented high values for Ni (217.33 mg/kg) and for As (60.56 mg/kg). The Zn values were homogeneous, except for sample 8, which showed a value considerably lower than the rest. On the contrary, sample 8 presented the highest value of Fe ($3.11 \times 10^4$ mg/kg). The Cr values for samples 1 and 5 were much higher than for the rest (383.00 and 127.50 mg/kg, respectively). Notwithstanding the disparity in the resulting values, no significant differences ($p = 0.8427$) were found regarding the heavy metal content among wells, despite the different confinement ages of the USW.

**Table 4.** Heavy metals and arsenic present in the USW from the closed dump (mg/kg).

| Well | Pb | Cu | Ni | Zn | Cr | Fe | As |
|---|---|---|---|---|---|---|---|
| 1 | 47.17 | 4.17 | 42.00 | 209.33 | 383.00 | $2.47 \times 10^4$ | 8.83 |
| 2 | 42.67 | 7.67 | 54.75 | 365.17 | 0.00 | $2.33 \times 10^4$ | 20.00 |
| 3 | 89.00 | 744.17 | 79.00 | 217.33 | 13.67 | $1.83 \times 10^4$ | 60.56 |
| 4 | 55.33 | 0.00 | 45.00 | 116.17 | 0.00 | $1.50 \times 10^4$ | 18.55 |
| 5 | 56.00 | 92.17 | 53.50 | 173.17 | 127.50 | $1.85 \times 10^4$ | 65.04 |
| 6 | 149.67 | 8.83 | 53.00 | 408.17 | 3.67 | $2.58 \times 10^4$ | 22.76 |
| 7 | 108.67 | 141.00 | 48.00 | 165.00 | 0.00 | $2.84 \times 10^4$ | 95.85 |
| 8 | 47.83 | 21.50 | 57.00 | 54.33 | 0.00 | $3.11 \times 10^4$ | 20.75 |
| Average | 74.54 | 127.44 | 54.03 | 213.58 | 57.19 | $2.31 \times 10^4$ | 39.04 |

## 4. Discussion

The degradation rate of the wastes and the physical-chemical characterization allowed the evaluation of handling of the dump during its operation stage, as well as the degradation behavior of the confined USW, and the effectiveness of the site closure measures.

The results showed a low moisture content of ~30%, still above that reported by [23] (15%) or by [24] (10%) as minimum values to favor USW optimal degradation. The degradation capacity is directly related to the moisture content of the samples. Therefore, these values of moisture are sufficient for 82% of the samples to show very fast decomposition because they are mainly composed of food wastes. However, higher moisture in the samples allows a better degradation of the residues since the optimal conditions for the establishment of the microbial consortia that degrade the wastes exist.

The failure to achieve optimal conditions for the establishment of organic matter degrading microorganisms delays the stabilization of the USW confined in the closed dump. In addition, the high holocellulose/lignin ratio of lignocellulosic compounds found in the 13% of the samples is also a delaying factor in stabilization of organic matter [25]. Still, we are aware that a larger number of samples should be analyzed to relate these results to the influence of the USW stabilization and confinement time.

Likewise, basic pH values indicate that acetogenic anaerobic bacteria would not be in their optimal environment since, according to [26], they develop optimally at a pH close to neutrality and are sensitive to pH variations. Despite this, the bacteria of these USW samples are active since they are totally inhibited at a pH below 6.0, which would be reflected in an accumulation of organic acids.

The deficiencies in waste stabilization are also directly related to the closure of the dump, whose work basically consisted of stabilizing the slopes and covering the USW with soil. Therefore, these deficiencies are probably not allowing the achievement of optimal conditions for the establishment of organic matter degrading microorganisms; which delays the stabilization of the USW confined in the closed dump.

Besides, moisture would also be affected by the closure measures. Previous studies in the study area [9] reported differences in the results obtained during the rainy and the dry season. Consequently, these measures also do not control the entry and dispersion of rainwater. In Morelia, precipitation increases 10 times during the rainy season, implying a considerable increase of the moisture values, as therefore in the amount of produced leachate. An increase in the amount of leachate generated would involve a greater dispersion of the pollutants they contain to surrounding soils and groundwater [27].

The results showed that five samples exceeded the maximum permissible level (MPL) for arsenic according to the national standard [28] about contaminated soils (Table 5). Although, these results should be reviewed with caution, because there is no other comparison parameter in Mexico for solid samples besides the MPL for sludge and biosolids, which were not exceeded. It highlights the urgency of legislation to regulate heavy metals

and arsenic in USW, in order to control the disposal of HW in landfills that are not designed and prepared for this purpose.

**Table 5.** Maximum permissible limits for metals and arsenic established in the Official Mexican Norms.

| Chemical Constituent | Sludge and Biosolids (mg/kg) | Contaminated Soils ** (mg/kg) |
|---|---|---|
| Pb | 300 | 400 |
| Cu | 1500 | NI * |
| Ni | 420 | 1600 |
| Zn | 2800 | NI * |
| Cr | 1200 | NI * |
| Fe | NI * | NI * |
| As | 41 | 22 |

** Total Reference Pollutant Concentrations ($PR_T$) by type of land use. NI *: Not included in the Mexican law. Modified from: NOM-004-SEMARNAT-2002 [29] anNOM-147-SEMARNAT/SSA1-2004 [28].

Besides the arsenic, we found that a worse scenario stands for copper, zinc and iron. For these pollutants, there are no reference values with which to establish whether the reported concentrations represent a danger to human and environmental health.

However, the heterogeneity in the values of the heavy metals and arsenic is due to the variability of the type of waste and the difference in the time of waste confinement; for which the Kruskal–Wallis test indicated a significant difference ($p = 0.01$), confirming the differences in the degradation state of the wastes of the different opencast wells.

Despite the fact that our findings are based on solid samples, this is in good agreement with the results of [9] in the same study area. Their geochemical analysis of groundwater from Morelia's municipal aquifer showed high concentration of heavy metals, exceeding the standards for drinking water of the World Health Organization. Subsequently, they analyzed samples of leachates during the rainy and dry seasons. Their results in leachates showed concentrations of Pb 1102 mg/L, Cu 2403 mg/L, Ni 10,678 mg/L, Cr 47,731 mg/L, and As 0.302 mg/L.

The results suggest that the possible percolation of leachates from the dump appears to be the most reasonable source of metal pollutants in the groundwater. The authors also explained that one of the factors in the production of leachate, and therefore dispersion of pollutants, was the joint deposition of organic and inorganic waste. In this regard, our study suggests that another major factor is the disposal of HW together with USW.

Many researchers, such as [30], have found high levels of heavy metals in the leachate as Fe, Pb, and Cd and a relatively small proportion of Zn, Cr, and Cu. Hence, our research stands out in that water pollution is a crucial problem that closed USW sites face due to dispersion of pollutants from leachates. The leachates would favor the solubility of the toxic components of USW due to its role as a catalyst for the degradation processes of hydrolysis, and dissolution of toxic components of organic and inorganic matter [31].

The aforementioned highlights the need for the implementation of monitoring to ensure the control of leachates produced at these sites during their operation and after closure. The situation is even more worrying in developing countries where research efforts towards monitoring the environment have not received the desired attention by stakeholders [1,2]. Furthermore, the retention of the pollutants in solution in the USW matrix is not assured due to the poor biodegradation of the organic matter in the waste, the lack of a liner, the absence of a leachates collection system and the poor coverage of the site [32].

## 5. Conclusions

The findings of this study indicate the presence of heavy metals and arsenic in the USW. These pollutants confirm that the solid wastes confined at the study site are a potential source of contamination.

Our research stressed that the HW confined clandestinely together with USW during 20 years reveal the illegality of the operation of the closed dump in Morelia, as well as the deficiencies in management during its operation and its later closure.

This work demonstrates the non-compliance with the Environmental Legislation regarding the USW disposal, the safe and adequate confinement. This study also underlines the importance of including the implementation of operational practices to control and/or avoid escape of those pollutants, likely from leachates which could reach the streams and/or water channels, affecting the quality of the water for human consumption.

Although our work includes only the analysis of one site, we provided further evidence that makes it urgent to review the methodologies for the USW disposal and the operation of landfills in developing countries. Further work needs to increase the number of solid samples and analyses of leachates and groundwater samples.

Despite the great environmental and health impact due to the poor management of landfills, in many developing countries, they still continue to be the main option for the treatment of USW. Studies of this type are therefore crucial for decision-makers and territorial planning.

**Author Contributions:** Conceptualization, O.B.-D. and M.L.G.-A.; methodology, O.B.-D.; validation, G.D.-V., R.A.-C.-V. and I.I.-A.; formal analysis, O.B.-D.; investigation, O.B.-D.; resources, O.B.-D.; data curation, G.D.-V., R.A.-C.-V. and I.I.-A.; writing—original draft preparation, O.B.-D. and M.L.G.-A.; writing—review and editing, O.B.-D. and M.L.G.-A.; visualization, G.D.-V., R.A.-C.-V. and I.I.-A.; supervision, O.B.-D. and M.L.G.-A.; project administration, O.B.-D.; funding acquisition, O.B.-D. All authors have read and agreed to the published version of the manuscript.

**Funding:** This work was funded by the National Council for Science and Technology of Mexico through Project Grant No. 62100 and the Co-ordination of Scientific Research of the Universidad Michoacana de San Nicolás de Hidalgo through Project Grant No. 5.9.

**Data Availability Statement:** The data presented in this study are available on request from the corresponding author.

**Acknowledgments:** The authors would like to thank Y.P. Meza-Cisneros for the laboratory analyses.

**Conflicts of Interest:** The authors declare no conflict of interest.

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
