# Peer review of "Hazardous Solid Waste Confined in Closed Dump of Morelia: An Urgent Environmental Liability to Attend in Developing Countries"

_sustainability, doi:10.3390/su13052557_

Round 1

Reviewer 1 Report

Review of  Sustainability - 1106873

“Hazardous solid waste confined in closed dump sites: an urgent  environmental liability to attend”

The aim of this study, as reported in the abstract by the authors, seems to be the determination of  the presence of heavy metals and arsenic in the confined wastes of a closed dump, in order to diagnose  the affectation from the contaminants. This aim is also confirmed at lines 88-90: “The aim of this work is to demonstrate the presence of heavy metals in the USW,  clandestinely confined together with hazardous waste”.

This manuscript is not a scientific work and cannot be accepted in a scientific journal as a research article in this actual version.

In detail:

  • The manuscript seems to be the transformation of a technical report about some (few) analysis on solid samples collected into a landfill.
  • In section Results, Table 1 refers to a general scheme of possible sources, Table 2 only describes the percentage of degradability of the organic fraction; Table 3 reports chemical-physical data; Table 4 reports data already collected in a technical report by other people about HM concentrations and Table 5 is a summary of the law limits imposed in Mexico.
  • No research contribute by the authors can be observed.

Moreover:

  1. the whole manuscript does not adopt  a rigorous scientific approach in describing and discussing the experimental tests. It is very difficult for the reader to understand where the analysis are carried out and to what environmental matrix they refer. The authors say they analyzed solids (waste). Have they analyzed samples of the landfill leachate and/or groundwater? If not, how can they verify contamination of groundwater?
  2. Table 5 shows the limits in different environmental matrices. What is the meaning of Table 5 if there are no analysis to compare? In lines 251-252 the authors discuss about leachates but do not report data. What is the meaning of this sentence: “The presence of heavy metals and arsenic in the confined residues indicates their  presence also in the leachates that are generated”?
  3. The whole work is not very clear. If the analysis only refer to solid samples and the landfill has not a liner (line 65) it is obvious that HM can pass into leachate and reach groundwater. So, this is not a source identification problem ( i.e. authors did not quantitatively correlate HM to specific types of HW that can be produced by specific industries or activities), nor a hazard identification problem, nor a risk assessment problem.
  4. The only conclusion of this work seems to be: “High concentration of HW have been measured inside samples of waste collected into a landfill. The landfill is closed and does not have the liner. The site is a possible hazard for population”.
  5. The section Discussion must be rewritten since a lot of statements are not supported by the very few data reported in the MS.

If the authors decide to prepare a completely new work about this topic, they must collect or report data about leachate and groundwater to discuss this “urgent environmental liability” (title of the paper). This means that new monitoring data must be collected, in order to correlate the content of HM found in the solid waste and the possible HM in groundwater (correlation by by infiltration models, advection-dispersion models). They  can also use data reported by other researchers.

Some detailed comments:

  1. The description of the landfill is confusing. The authors should say immediately that this is a close landfill without a liner.
  2. In lines 35-36 they refer to leachate. Do the limits are on concentrations of HM in leachate (do limits NOM-127-66A1-1994 correspond to leachate released from a municipal solid waste?). In lines 54-55 they say “USW samples”. Are these solid samples? In lines 120-121 it seems that solid samples of USW are collected from the site. Also in lines 126-140 it seems that only solid samples are analyzed.
  3. Table 1: are these “General sources identified” or “possible general sources”? If this is the result of an identification procedure, where are the methods and statistics? (probability distribution, etc.)
  4. Table 5: please refer to the environmental compartment (matrix) and not to the name/number of the NOM.
  5. The title is too generic.

Reviewer 2 Report

The article is interesting but requires some refinement. The information contained in the abstract does not correlate to the information contained in the text. Where are the results of the piezometer leachate analyzed?. Where are the soil analysis results?. Wells are mentioned in the abstract. Are they wells or opencast ?. The results of the analyzes in Table 4 are the author's results?.  Who and with what instrument performed the analyzes, eg. Fe to 7 significant numbers?. Some of the literature is old. Needs to be standardized. The high temperature in the landfill indicates a turbulent process of mineralization. For the characteristics of the leaching of the metals, sequential chemical extraction of metals is better.

Round 2

Reviewer 1 Report

The authors improved the overall quality of their manuscript. Now their analysis and work is clear. Moreover, also methodology is well described.

Anyway, my doubts about the the scientific contribute of this work still remain.

Even if the problem of the not authorized confinement of HW in landfills devoted to MSW collection is an important issue (not only in developing countries) a work that describe the finding of HM in a specific single landfill cannot be considered a scientific work.

Reviewer 2 Report

I only have remarked on table 5. Sludge biolosids or biosolids? mg/Kg or mg/kg?

Round 3

Reviewer 1 Report

The manuscript was not changed (equal to version v2). My comments remain unchanged.
